# Monitoring Electrochemical Reactions in Situ with Low Field NMR: A Mini-Review

**Bruna Ferreira Gomes** [1,†] , **Carlos Manuel Silva Lobo** [1,†] **and Luiz Alberto Colnago** [2,*,†]

[1]  Instituto de Química de São Carlos, Universidade de São Paulo, São Carlos 13566-590, Brazil;
    bruna.fg.lobo@gmail.com (B.F.G.); carlos.manuel.s.lobo@gmail.com (C.M.S.L.)
[2]  Embrapa Instrumentação, São Carlos 13560-970, Brazil
[*]  Correspondence: luiz.colnago@embrapa.br; Tel.: +55-16-2107-2821
[†]  All authors contributed equally to this work.

**Abstract:** The number of applications of time domain NMR using low-field spectrometers in research and development has been steadily increasing in recent years with applications ranging from quality control of industrial products to the study of physical and chemical properties of a wide array of solid and liquid samples to, most recently, electrochemical studies. In this mini-review we summarize the progress that has been achieved in the coupling between time domain NMR (using low-field spectrometers) and electrochemistry and how the challenges that this coupling poses have been overcome over the years. We also highlight the effect that the static magnetic field of the NMR spectrometer has on the electrochemical systems.

**Keywords:** time domain NMR; electrochemistry; EC-NMR

## 1. Introduction

The uses of low-field time domain NMR in research and development have been steadily increasing over the years [1–6]. Nowadays, it is used for several purposes, such as the quality analysis of industrial products as well as the study of physical and chemical properties of a wide array of samples, both solid and liquid, such as vegetable oil, fuels, reservoir rocks, foods and biological tissues [1–6]. The instruments used for these analyses are normally based on permanent magnets which produce a magnetic field intensity usually below 0.6 T (25 MHz for $^1$H) with a low homogeneity (above 100 ppm) which means that the relevant information can only be extracted from the intensity of the free induction decay signal (FID), the spin echo or by measuring the relaxation times $T_1$ and/or $T_2$ or the diffusion coefficient of the sample. In this review we show the application of time domain NMR (TD NMR—also known as low-field NMR) to an increasingly popular research field which is the in situ monitoring of electrochemical reactions (EC-NMR) [7–9]. The first published paper which detailed the use of TD NMR for such an application was published as recently as 2010 [10] (which is a 40 year delay from the first paper detailing the use of high-field NMR for the same application [11]). This work showed that TD NMR is suitable for monitoring the electrodeposition reaction of copper complexes using the transversal relaxation time, $T_2$. Since then, TD NMR has been shown to be suitable for monitoring the electrodeposition of copper ions using both H-type magnets as well as unilateral magnets. Furthermore, it has been observed that reactions performed in situ show a higher reaction rate than those performed ex situ due to a phenomenon known as magnetoelectrolysis [12,13] which occurs as a result of the interaction between the magnetic field and the ion flow present during the experiments. This effect increases the mass transport within the solution during the experiment. In this review, we show the methods used to measure the concentration of paramagnetic ions in solution using TD NMR, the challenges of in situ EC-NMR and the solutions to those challenges (whenever they

exist) as well as the influence of the magnetic field from the spectrometer on the magnetoelectrolysis. The use of inductively coupled coils to improve the signal-to-noise ratio of the NMR signal during in situ electrochemical experiments is also shown here.

## 2. TD NMR for the Quantification of Paramagnetic Ions

In 1946, Bloch, Hansen and Packard reported that $T_2$ of the sample is dependent upon the concentration of the paramagnetic ions [14]. This was later elaborated on by several scientists who reported that $T_2$ is inversely proportional to the concentration of paramagnetic ions in the solution [15–17]. The relaxation of the solvent due to the effect of the paramagnetic ions in the solution was first explained by Bloembergen, Purcell and Pound in 1948 [15]. They demonstrated that the longitudinal relaxation rate $(1/T_1)$ is dependent on the gyromagnetic ratio of hydrogen $(\gamma_p)$, effective magnetic moment $(\mu_{eff})$, ion density $(N_{ion})$, viscosity $(\eta)$, Boltzmann's constant$(k)$ and absolute temperature $(T)$, as shown in Equation (1).

$$\frac{1}{T_1} = \frac{12\pi^2 \gamma_p^2 \eta N_{ion} \mu_{eff}^2}{5kT} \tag{1}$$

In the 1970s, Schlüter and Weiss [18–20] indirectly quantified the concentration of paramagnetic ions in solution by measuring the relaxation rate $(1/T_1)$ of the solution and taking advantage of the inverse correlation between both properties. Since measuring $T_1$ using the conventional inversion-recovery pulse sequence is a long process, and since Bloembergen, Purcell and Pound reported that $(1/T_2)$ is also proportional to the concentration of paramagnetic ions in solution it is possible to indirectly quantify the concentration of such ions using $T_2$, which can be measured in a fast and simple way using the CPMG pulse sequence. Figure 1 shows how the transversal relaxation rate $(1/T_2)$ varies linearly with the concentration of several paramagnetic ions [21].

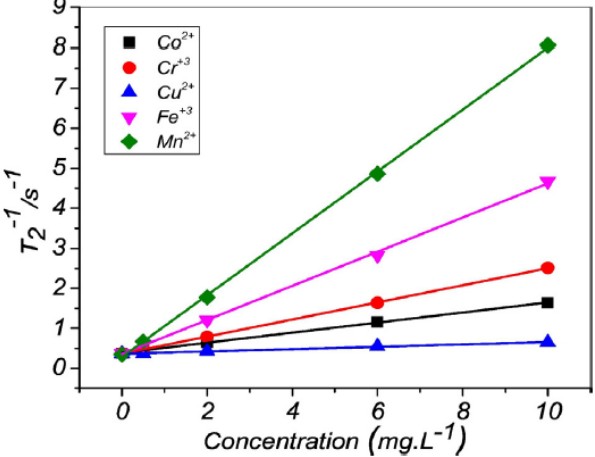

**Figure 1.** Variation of transverse relaxation rate $(T_2^{-1})$ of the water versus the concentration of $Co^{2+}$, $Cr^{3+}$, $Cu^{2+}$, $Fe^{3+}$ and $Mn^{2+}$ in the solutions. Reprinted from Publication [21], Copyright (2018), with permission from Elsevier.

In 2016 Gomes et al. determined the limits of detection and quantification (LOD and LOQ, respectively) of two low-field NMR equipments. One based on a H-type magnet (0.23 T —9.8 MHz for $^1$H) and the other using a unilateral magnet (0.33 T at 3 mm above the magnet surface—14.1 MHz for $^1$H) using different paramagnetic ions ($Ni^{2+}$, $Cu^{2+}$, $Cr^{3+}$ and $Mn^{2+}$). These values were compared to the LOD and LOQ of a Uv/Vis spectrometer for the same ions, with the exception of $Mn^{2+}$ which does not significantly absorb radiation in the Uv/Visible spectrum to be quantified. The authors concluded that the LOD and LOQ using the unilateral sensor was the highest, followed by those obtained by the Uv/Vis spectrometer, which also has the drawback of not being able to quantify ions that do not

absorb radiation in this region (as is the case with $Mn^{2+}$ ions). The NMR spectrometer with the H-type magnet showed the lowest LOD and LOQ. The average LOD and LOQ of all the paramagnetic ions tested and on each of the different magnets are summarized in Table 1 [22].

**Table 1.** Average limits of detection and quantification (LOD and LOQ) of the different spectrometers for the tested paramagnetic ions [22].

| Magnet Type | LOD (mol·L$^{-1}$) | LOQ (mol·L$^{-1}$) |
|:---:|:---:|:---:|
| H-type | $4 \times 10^{-6}$ | $2 \times 10^{-5}$ |
| Unilateral | $1 \times 10^{-3}$ | $5 \times 10^{-3}$ |
| Uv/Vis | $6 \times 10^{-4}$ | $2 \times 10^{-3}$ |

The linear relationship between the relaxation rate ($1/T_2$) and the concentration of paramagnetic ions has been exploited to quantify paramagnetic ions in solutions, to calculate $K_{SP}$ values [23], to study coordination complexes with paramagnetic ions as ligands [24] and also to monitor electrochemical reactions in situ [8,9].

## 3. TD NMR Coupled to Electrochemistry

Electrochemical measurements are quite limited when it comes to discriminating how much of each compound involved in the reaction is consumed or produced, since they measure the total current that flows through the system, which includes not only the target reaction but also parallel reactions such as gas evolution. Thus, it is not possible to exactly determine how much of the current has been used to produce a certain compound based on these measurements alone. It is for this reason that coupling NMR to electrochemical methods is useful, as NMR (more specifically high-field NMR) can provide information regarding the presence and concentration of the different soluble molecules involved in the reaction. On the other hand, TD NMR can be used to follow the consumption of paramagnetic ions in the solution through measurements of either $T_1$ or $T_2$.

Taking advantage of this fact, in 2010 Barbosa et al. [10] used an electrochemical cell with a height of 6 cm and 3 cm of diameter (Figure 2) in a 2.1 T superconductor magnetic resonance imaging system to follow in situ the electrodeposition reaction of copper ions in a solution containing sorbitol as a complexing agent. However, they could not observe a substantial difference in the values of $T_2$, which could be linked to the large cell volume.

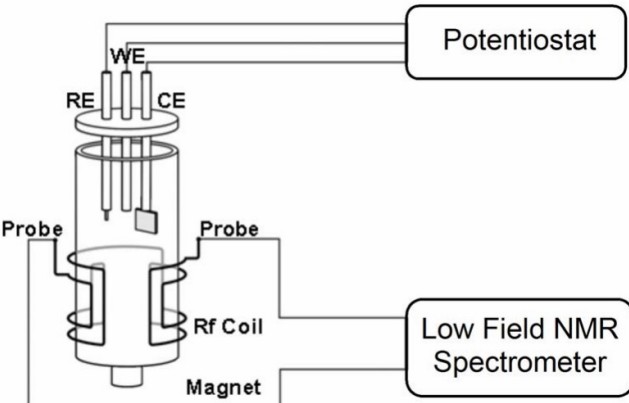

**Figure 2.** Schematic diagram of the EC-NMR cell used by Barbosa et al. WE, CE and RE refer to the working, counter and reference electrodes, respectively. Reproduced with permission from [10]. Copyright (2010), The Electrochemical Society.

Introducing an electrochemical cell in the NMR detection region does bring with it some challenges. First and foremost, introducing metal electrodes in the detection region of the NMR spectrometer reduces the signal-to-noise ratio of the NMR experiments by introducing noise and by reducing the intensity of the signal due to distortions caused to the static magnetic field, which reduces the homogeneity of the magnetic field. Typical ways to deal with these problems include using radio-frequency filters (chokes), the use of non-metallic electrodes and even the use of metallic thin films [25].

In 2012 Nunes et al. [7] published the first article showing that it is possible to monitor the electrodeposition reaction of $Cu^{2+}$ ions by coupling the electrochemical experiment to a low-field bench-top NMR spectrometer (0.23 T). They measured the $T_2$ of the solution using the CPMG pulse sequence and were able to track the consumption of the copper ions in real-time (Figure 3) despite the fact that the noise in the NMR signals was increased. The authors placed the electrodes in the NMR detection region and made no use of radio-frequency filters. However, they did isolate the cables connecting the electrochemical cell to the potentiostat to reduce the interference that the electrodes cause on the NMR signal.

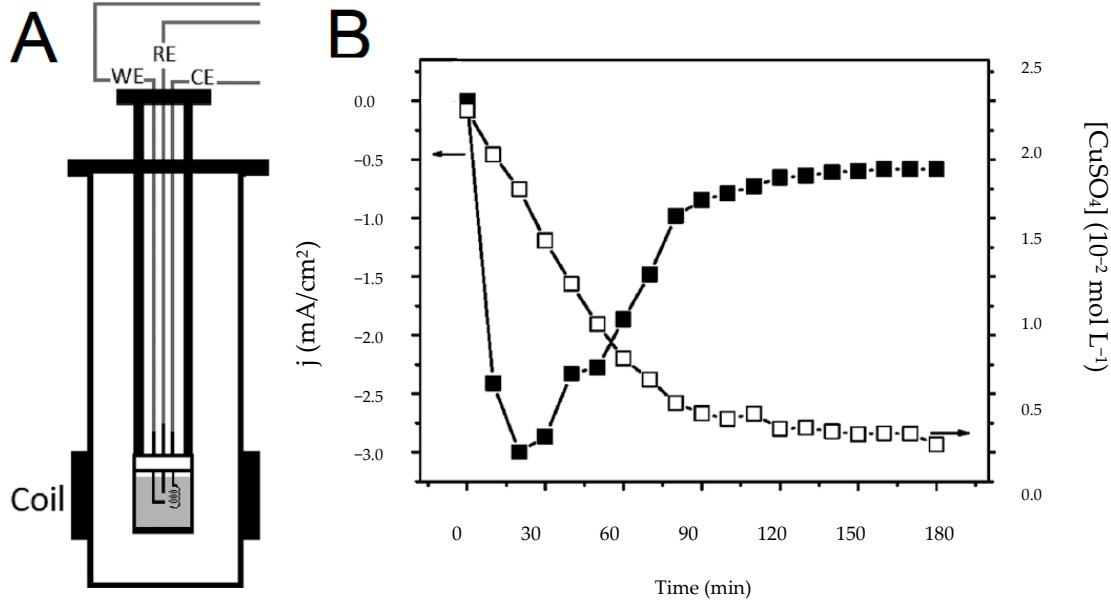

**Figure 3.** (**A**)The electrodes of the cell were as follows: working electrode (WE) and counter electrode (CE), platinum; reference electrode (RE), silver. (**B**) Variation of the $Cu^{2+}$ concentration (□) and the potentiostat current (■) during the electrodeposition reaction. Adapted with permission from [7]. Copyright (2012) American Chemical Society.

In 2018, Lobo et al. [26] showed that it is possible to increase the NMR signal-to-noise ratio of the TD NMR experiments performed during in situ electrochemical experiments by using a secondary coil which is wrapped directly on the electrochemical cell. The system that was used consisted of an electrochemical cell built into a 5 mm standard NMR tube which contained a Pt wire as CE and an Ag/AgCl wire as RE positioned above the secondary electrode and a WE consisting of a Pt disc placed at the bottom of the tube (Figure 4A). The developed setup achieved an increase of 20 times in the NMR signal intensity and was used to successfully follow the electrodeposition reaction of cupric ions at two distinct potentials (vs. Ag/AgCl): −1 and −4 V (Figure 4B).

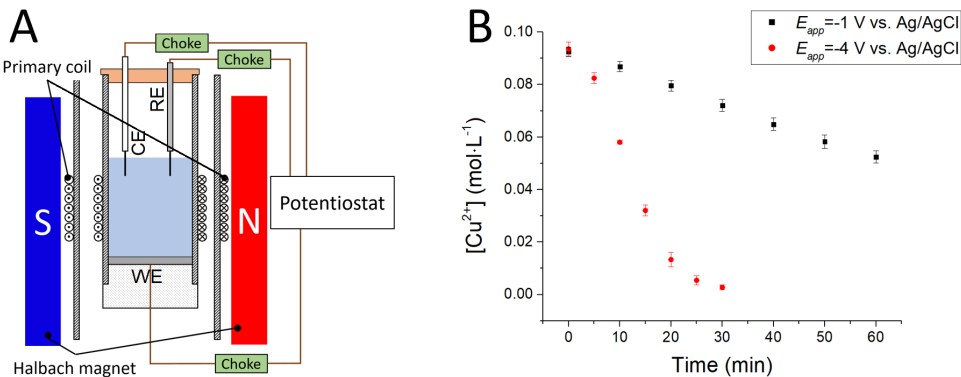

**Figure 4.** (**A**) Experimental setup using an inductively coupled coil to enhance the NMR signal-to-noise ratio. (**B**) Cupric ion concentration over time. Reactions performed at –1 and –4 V (vs. Ag/AgCl) are represented by black squares and red circles, respectively. Adapted from [26], copyright (2018), with permission from Elsevier.

## 4. TD NMR for the Study of Magnetoelectrolysis

In 2014, Gomes et al. [8] used a similar system to that used by Nunes et al. in 2012 and observed that the reaction rate of the $Cu^{2+}$ electrodeposition reaction monitored in situ by TD NMR was faster than the same reaction performed ex situ (Figure 5). The results showed that after 1 h of reaction 40% of the copper ions had been deposited on the electrode surface when the reaction was performed in situ, which is a 48% increase in copper deposition when compared to the 27% that was deposited when the reaction was performed ex situ (Figure 5). The increase in the reaction rate was associated with the magnetoelectrolysis effect, which is known to stir the solution, and increases the quantity of ions that reach the electrode surface.

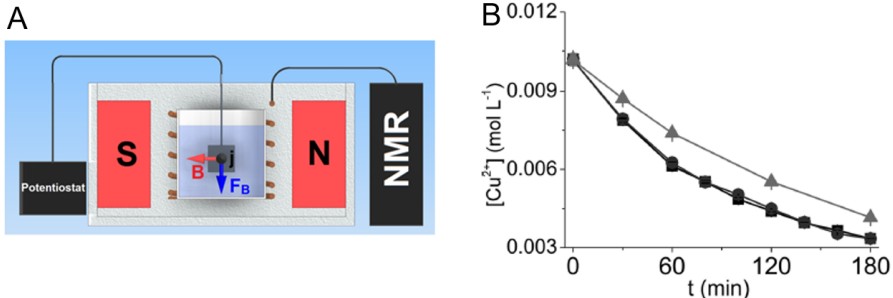

**Figure 5.** (**A**) Illustration of the electrochemical cell inserted in the NMR relaxometer. (**B**) Variation of $Cu^{2+}$ concentration as a function of electrolysis time performed in the electrochemical cell recorded at 23 °C under different configurations: (■) $\mathbf{B}\perp\mathbf{j}$, (●) $\mathbf{B}\|\mathbf{j}$, and (▲) $\mathbf{B} = 0$. Solution: 0.1 mol·$L^{-1}$ $Na_2SO_4$ electrolyte containing 0.01 mol·$L^{-1}$ $CuSO_4$. Conditions: $E_{applied}$ = –0.4 V vs. Ag/AgCl (3 mol·$L^{-1}$ KCl). The $Cu^{2+}$ concentration data were determined by CPMG pulse sequence. The standard error is represented by the error bars (*n* = 3). Adapted with permission from [8]. Copyright (2014) American Chemical Society.

The magnetoelectrolysis effect is proportional to both the magnetic field intensity, **B**, and the current density, **j**, that reaches the electrode surface. The most important force that results from the interaction of these quantities is the magnetic field force density, $\mathbf{F_B}$, which is given by Equation (2).

$$\mathbf{F_B} = \mathbf{j} \times \mathbf{B} \tag{2}$$

where **j** has units of A $m^{-2}$, **B** is given in T and $\mathbf{F_B}$ has units N $m^{-3}$. From Equation (2) we can see that $\mathbf{F_B}$ will be maximum when **j** and **B** are perpendicular to each other and null when they are parallel. In practice, however, even if at a macroscopic level **j** and **B** are parallel, the electrode surface as well as

its edges create local distortions to the electric field which results in there being regions where **j** and **B** are no longer parallel and thus $\mathbf{F_B}$ will be minimum, but not zero, as explained Monzon et al. in 2015 [13], who call this particular effect micro-magnetohydrodynamic effect.

Gomes et al. (2014) [8] also discuss the reason for why the reaction rate remains unaltered despite an apparent parallel orientation between **B** and **j** (Figure 5). They explain this result in terms of the effect of the edges of the electrode and gas bubbles that form on its surface during the reaction, which locally alter the electric field, therefore preventing $\mathbf{F_B}$ from being zero.

Hinds et al. in 2001 [12] explained that the increase of an electrochemical reaction's velocity due to $\mathbf{F_B}$ is due to the fact that this force agitates the solution, which leads to more molecules reaching the surface of the working electrode and, therefore, reducing the thickness of the electric double-layer (Figure 6). They also explained that this increase only occurs in those reactions which are limited by mass transport, since the magnetic agitation caused by $\mathbf{F_B}$ may be compared to a mechanical agitation, which homogenizes the solution. Aogaki et al. demonstrated that the increase in the conversion rate (determined through the measurement of the electric current that flows during the experiment) is proportional to **B** [27].

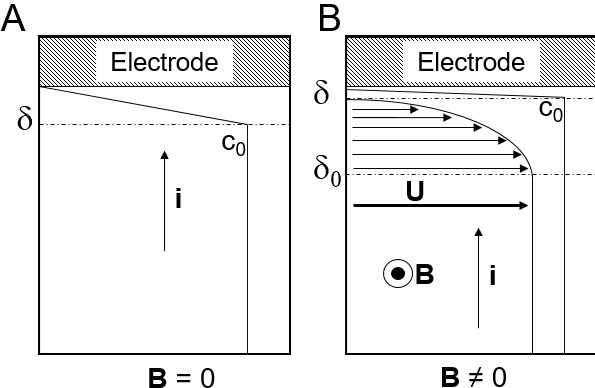

**Figure 6.** (**A**) Electric double-layer thickness ($\delta$) during electrochemical experiments performed in the absence of an external magnetic field, **B**. The bulk concentration of ionic species is $c_0$. (**B**) Electric double-layer thickness when $\mathbf{B} \neq 0$, which induces a tangential velocity component, **U**, in the bulk solution which in turn creates a hydrodynamic boundary layer with thickness $\delta_0$. **i** is the current flowing in the system. Adapted with permission from [12]. Copyright (2001) American Chemical Society.

Another force that may play an important role in the magnetoelectrolysis effect is the force of the magnetic field gradient, also known as Kelvin's force, $\mathbf{F_{\nabla B}}$, which is more influent the less homogeneous the magnetic field is, as shown by Equation (3).

$$\mathbf{F_{\nabla B}} = \frac{1}{2\mu_0} c \chi_m 2(\mathbf{B} \cdot \nabla)\mathbf{B} \tag{3}$$

where $\mu_0$ is the magnetic permeability of free space in units of T m A$^{-1}$, $c$ is the concentration of paramagnetic species in solution given in mol m$^{-3}$, $\chi_m$ is the molar magnetic susceptibility, **B** is the external magnetic field in T, $\nabla$ is the Del operator and $\mathbf{F_{\nabla B}}$ is given in units of N m$^{-3}$. This force can alter the morphology of metallic deposits as well as contribute to the magnetic agitation of the solution.

In 2015, Gomes et al. [9] again showed that it is possible to follow the electrochemical deposition of copper ions, this time using a unilateral, homebuilt, sensor. To do this, however, a Faraday cage was built around the system to isolate it from ambient radio-frequencies which are driven into the NMR detection region by the electrodes and completely destroy the NMR signal. The cage consisted of a simple box whose inner walls were lined with aluminum foil. The reactions performed in situ (they again tested both $\mathbf{B} \perp \mathbf{j}$ and $\mathbf{B} \parallel \mathbf{j}$) showed that after the 3h reaction 40% of copper had been

deposited while only 12% had been deposited during the ex situ reaction, the former value being independent of the orientation between **j** and **B**.

The authors also investigated and estimated the magnetic forces at play during the in situ experiments and attributed the higher deposition rate of the in situ reactions to the effects of $F_B$ and $F_{\nabla B}$, the latter being quite important as the magnet produces a highly inhomogeneous magnetic field (the gradient was estimated to be 6 T m$^{-1}$). Just as with the work from 2014, the reaction rate remained the same whether **j** and **B** were parallel or not (Figure 7), for the same reasons.

The works from Gomes et al. from 2014 and 2015 highlight the importance of taking the effect of the magnetic field into account whenever EC-NMR experiments are performed in situ, since the reaction rate may differ from the rate observed in an ex situ reaction and this difference is proportional to the intensity of the applied magnetic field as shown in Equation (2). Thus, before initiating any in situ EC-NMR experiments the influence of the external magnetic field must be assessed to avoid inaccurate conclusions. It is worth pointing out that the effect of the magnetic field is especially evident on reactions that are limited by mass-transport because it has the effect of reducing the thickness of the electric double-layer due to magnetic stirring.

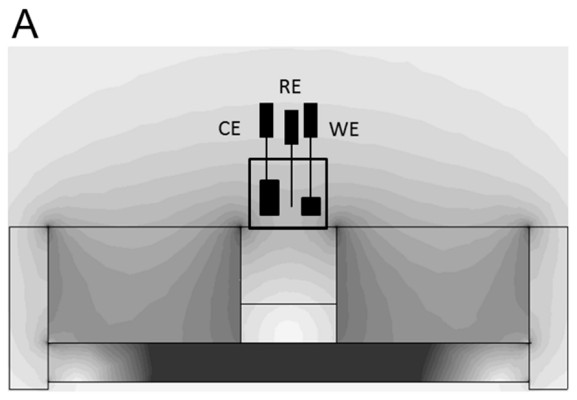 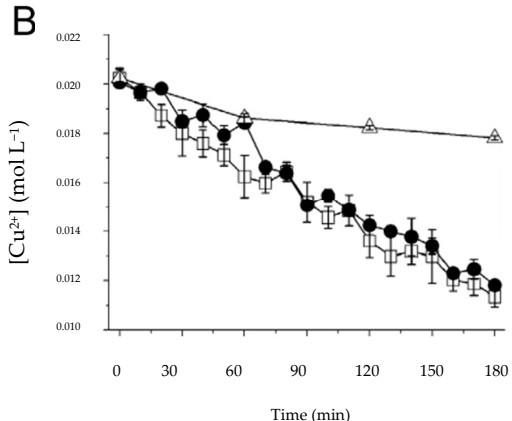

**Figure 7.** (**A**) Simulated magnetic field profile of the home-built unilateral NMR sensor. The simulation procedure was performed without the electrochemical cell on unilateral probe. The gray color scale represents the magnetic field from 0.05 to 1.0 T. Each color scale is equivalent to 0.05 T. (**B**) Variation of the $Cu^{2+}$ concentration during the electrodeposition reaction measured by CPMG pulse sequence under different configurations: **B** = 0 ($\triangle$), **B**$\perp$**j** ($\square$) and **B**$\|$**j** ($\bullet$). $E_{applied}$ = −0.5 V vs. Ag wire. T = 25 ± 1 °C. The error bars were calculated from three replicates. Adapted from Publication [9], Copyright (2015), with permission from Elsevier.

## 5. Conclusions

In this mini-review it was demonstrated that low-field, TD NMR could be a useful method to study electrodeposition reactions of paramagnetic ions. It also shows that it is possible to study the effect of the magnetic field on the electrodeposition reaction rate (magnetoelectrolysis) through TD NMR. Therefore, this low field NMR spectrometer can be an important tool to study several galvanization processes, not only with $Cu^{2+}$, but also with other paramagnetic ions such as $Cr^{3+}$, $Ni^{2+}$, that have many industrial applications.

**Author Contributions:** All the authors contributed equally to the preparation of the manuscript.

**Funding:** This work was supported by CNPq grant number 161555/2015-2, FAPESP grants number 2012/22281-9, 2017/12864-0 and 2016/01537-6 (BEPE) and DAAD grant number 91635214.

**Conflicts of Interest:** The authors declare no conflict of interest.

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
