# Peer review of "Monitoring Electrochemical Reactions in Situ with Low Field NMR: A Mini-Review"

_applsci, doi:10.3390/app9030498_

Round 1

Reviewer 1 Report

I enjoyed reading and recommend publishing the mini review by Gomes that briefly summarizes the current state of a research area that at present is relatively niche area, namely the use of low field NMR spectroscopy to monitor in-situ electrochemistry.

In terms of content I have only one significant comment.  It was not really clear to me what the advantage is of using NMR to monitor in-situ electrochemistry given a number of drawbacks. The paramagnetic species concentration is indirectly monitored through the effect on the water relaxation behavior and as I understand it only one paramagnetic species can therefore be monitored. Indeed, at present the available work on that subject seems to have focused on just copper solutions. By comparison, coulometry is a well-established method to monitor and control quantitatively electrodeposition. I believe it would be helpful to the reader if the authors more clearly point out the (potential) advantages of low field NMR spectroscopy in particular over coulometry.

Some editorial corrections:

Line 21 omit “when”

Line 26 “it has” not “it’s”

Line 30 there is something wrong/hard to read with the sentence starting with “The methods used to measure…” Perhaps two sentences were intended?

Line 39 us really “ionic” the key property here or do you mean paramagnetic ion concentration

Line 65 omit “so”

Line 66 this seems to be misleading, only species is monitored at a time such as just Cu2+ during electrodeposition

Line 149 …magnetic field as shown in equation 2.

Line 152  something wrong with the phrasing here concerning “seeing as it acts”  

Author Response

In terms of content I have only one significant comment.  It was not really clear to me what the advantage is of using NMR to monitor in-situ electrochemistry given a number of drawbacks. The paramagnetic species concentration is indirectly monitored through the effect on the water relaxation behavior and as I understand it only one paramagnetic species can therefore be monitored. Indeed, at present the available work on that subject seems to have focused on just copper solutions. By comparison, coulometry is a well-established method to monitor and control quantitatively electrodeposition. I believe it would be helpful to the reader if the authors more clearly point out the (potential) advantages of low field NMR spectroscopy in particular over coulometry.

Indeed this explanation was missing and we have explained in the text (line 70, in Section 3) that electrochemical measurements "measure the total current that flows through the system, which includes not only the target reaction but also parallel reactions such as gas evolution. Thus, it is not possible to exactly determine how much of the current has been used to produce a certain compound based on these measurements alone." On the other hand (high-field) NMR is able to distinguish between different compounds that are formed/consumed in the reaction. Furthermore, a comparison of the limits of detection and quantification (LOD and LOQ) of NMR and Uv/Vis (line 56, before Table 1) is shown which demonstrates that NMR possesses a better LOD and LOQ than Uv/Vis and that NMR can quantify certain species which do not absorb Uv/Vis radiation, therefore making them impossible to detect with this method.

Line 21 omit “when”

This has been done.

Line 26 “it has” not “it’s”

This has been corrected.

Line 30 there is something wrong/hard to read with the sentence starting with “The methods used to measure…” Perhaps two sentences were intended?

The sentence has been rewritten. It now reads: 

In this review, we show the methods used to measure the concentration of paramagnetic ions in solution using TD NMR, the challenges of in situ EC-NMR and the solutions to those challenges (whenever they exist) as well as the influence of the magnetic field from the spectrometer on the magnetoelectrolysis.

Line 39 us really “ionic” the key property here or do you mean paramagnetic ion concentration

The sentence has been corrected. It now reads:

T2 is inversely proportional to the concentration of paramagnetic ions in the solution.

Line 65 omit “so”

It has been removed.

Line 66 this seems to be misleading, only species is monitored at a time such as just Cu2+ during electrodeposition

This phrase actually refers to high-field NMR, which does have such capabilities. This has been explicitly stated in the sentence.

Line 149 …magnetic field as shown in equation 2.

This change has been made.

Line 152  something wrong with the phrasing here concerning “seeing as it acts”  

the sentence has been updated and now reads:

It is worth pointing out that the effect of the magnetic field is especially evident on reactions that are limited by mass-transport because it has the effect of reducing the thickness of the electric double-layer due to magnetic stirring. 

Reviewer 2 Report

The authors do a nice job of summarizing what appears to be mostly their own work in this application of NMR to monitor electrochemical reactions in situ. The paper is clearly written, and provides a nice entry point into the literature for those interested in the area.

Author Response

We thank the reviewer for taking the time to read and evaluate the article.